# Role of Changes in State of Bound Water and Tissue Stiffness in Development of Age-Related Diseases

**DOI:** 10.3390/polym12061362

**Published:** 2020-06-17

**Authors:** Garry Kerch

**Affiliations:** Faculty of Materials Science and Applied Chemistry, Riga Technical University, 1048 Riga, Latvia; garrykerch@inbox.lv

**Keywords:** bound water, hydration, stiffness, age-related diseases

## Abstract

An essential effect of environmental stiffness on biological processes in cells at present is generally accepted. An increase in arterial stiffness with advanced age has been reported in many publications. The aim of the present review is to summarize current information about possible chemical reactions and physical processes that lead to tissue stiffening and result in age-related diseases in order to find methods that can prevent or retard time-dependent tissue stiffening. The analysis of published data shows that bound water acts as a plasticizer of biological tissues, a decrease in bound water content results in an increase in biological tissue stiffness, and increased tissue stiffness leads to NF-kB activation and triggered actin polymerization—NF-kB activation is associated with age-related diseases. It can be suggested that changes in bound water content through changing tissue stiffness can affect cellular processes and the development of pathologies related to aging. Both age-related diseases and COVID-19 may be associated with tight-junction disruption and increased tissue stiffness and permeability.

## 1. Introduction

The physical, chemical, and biological behavior of macromolecules in biological tissues depends on the content and binding energy with water molecules. Proteins and other macromolecules in living tissue are functional provided that they contain water molecules as an integral part. Biological macromolecules are inactive in the absence of bound water. The changes in bound water and free water ratios have been associated with age-related and protein-conformational diseases and have an important role in time-dependent processes in biological tissues [1,2,3]. The content of tightly bound water in biological tissues decreases with advanced age [1,2]. Nuclear magnetic resonance (NMR) and differential scanning calorimetry (DSC) are methods that can determine the state of water. NMR and differential thermal analysis (DTA) studies demonstrated variation in water binding energy from tightly bound water molecules to loosely bound water molecules depending on the changes of moisture content in a polymer material. Commonly, three regions of proton molecular mobility represented by relaxation times can be observed using NMR. The relaxation time of free water is much higher if compared with the relaxation time of tightly bound water molecules. Otsuka and co-authors [4] suggested the application of near-infrared spectroscopy to study states of water. Absorbance spectra in the frequency range of 4500–5500 cm^−1^ were resolved into three peaks, which correspond to water molecules with different hydrogen bond states.

The behavior of unfrozen water below the equilibrium free water freezing temperature has been discussed by Wolfe and co-authors [5]. Free water freezes at 0 °C, but freezing point depression can be observed for loosely bound water molecules absorbed by polymers. Wolfe et al. suggested that the binding energy of water molecules depends on the distance from the hydrophilic surface or hydrophilic functional groups of macromolecules, and higher water freezing point depression is related to higher energy of interaction with the surface or macromolecule. The tightly bound unfreezable water molecules in the first and second molecular layers next to the surface are less mobile than the water molecules that are at a higher distance from the surface [5,6,7].

It has been concluded that water in close contact with macromolecules is no longer liquid but somewhat structured [8]. Bound water also controls the conformational changes in biological macromolecules.

The effect of living tissue plasticization by water on tissue stiffness is typically neglected in the current state of the art. The aim of this paper is to review mechanisms that demonstrate how changes in bound water content can lead to changes in tissue stiffness, promoting the aging process and triggering related pathologies.

## 2. Bound Water and Tissue Stiffness

### 2.1. Tissue Stiffness

Discher and co-authors reported in 2005 that cells respond to the rigidity of their planar substrate [9]. It has been further demonstrated that cellular processes in the three-dimensional matrix respond differently to microenvironment stiffness and, for example, in hyaluronic acid hydrogels, the differentiation behavior of human mesenchymal stem cells (hMSCs) depends on cellular traction. High degrees of cell spreading on stiffer surfaces favored osteogenesis, and lower degrees of cell spreading on softer surfaces favored adipogenesis [10]. Protein-conformational diseases and age-related diseases are associated with increasing tissue stiffness with increasing age [1,11,12]. Arterial stiffness was associated, not only with cardiovascular diseases, but also with neurodegenerative diseases, cognitive impairment, and vascular dementia [13,14,15]. Thus, the prevention of tissue stiffening is a therapeutic approach with the clinical potential to attenuate disease, which has been discussed in a recent review by Lampi and Reinhart-King [11]. It has also been suggested that antihypertensive drugs can decrease vascular stiffness and reduce the occurrence of cognitive impairment [15]. Increased extracellular matrix (ECM) stiffness was related to cancer, cardiovascular disease, diabetes, and other pathologies. ECM provides biomechanical cues that direct cell growth, migration, differentiation, and survival [16].

### 2.2. Tightly Bound Water

It is important to take into account that mechanical properties depend on the content and state of water bound to biological macromolecules [2,3]. The binding energy of water molecules to biological macromolecules decreases with aging, and tightly bound water transforms into loosely bound and free water. Tightly bound water is a plasticizer and the decrease in its content results in an increase in tissue stiffness. The decrease in bound water fraction in the extracellular matrix with advanced age was correlated with the loss of mechanical properties of the cortical bone in female and male BALB/c mice [17]. Strength and toughness of hydrated human bone decreases with decrease of bound water content, indicating the risk of fracture [18]. In bone, the quantity of water bound to collagen decreases with age [19]. Total body water (TBW) and intracellular water (ICW) decrease with age, but elderly individuals with higher ICW content have higher muscle strength and better functional performance [20]. An increase in the extracellular water (ECW) to ICW ratio in the upper legs determined using segmental bioelectrical impedance spectroscopy was associated with a decrease in muscle strength in older individuals [21]. Kim and co-authors demonstrated that high ECW/ICW ratio is a major risk indicator for all-cause mortality and cardiovascular diseases. A correlation between ECW/ICW with inflammation and pulse wave velocity was reported [22]. ECW/ICW also has been reported as a prognostic factor for survival in metastatic cancer [23]. A decrease in the ICW content with age proceeds with a higher rate than in the ECW content. Increase in ECW/ICW ratio accelerates after the age of 70 years [24].

Thus, it is very important to prevent or retard the transformation of intracellular tightly bound water into extracellular loosely bound and free water in order to mitigate an increase in tissue stiffness.

### 2.3. Advanced Glycation End Products

It has been shown that tissue stiffness increases as a result of tissue crosslinking by advanced glycation end products (AGEs). Dehydration and glycation may be considered as subsequent steps during the aging process [3,25]. A decrease in bound water content and increase in AGEs during aging were reported in the cortical bones of rats [26]. The accumulation of AGEs and reduction of water content in the ovine intervertebral disc was also observed [27]. Crosslinking of elastin and collagen by AGEs was associated with stiffening of the arterial wall with aging [12,16] and AGE cross-link breakers can be suggested as “de-stiffening” drugs [28,29].

A number of AGE inhibitors that are effective in decreasing AGE accumulation in tissues have been proposed, such as aminoguanidine, pyridoxamine, aspirin, ALT-711 (alagebrium), thiamine, tenilsetam (antidementia drug), and others [30,31,32,33,34,35,36].

### 2.4. De-Stiffening

#### 2.4.1. Polyphenols

Polyphenols have been suggested as inhibitors of AGEs [37,38,39,40,41,42]. Diabetes leads to glycocalyx degradation, and the maintenance or restoration of the integrity of the glycocalyx is an important therapeutic target. Polyphenols, such as resveratrol, quercetin, and (−)-epicatechin, can promote the swelling of the highly hydrated endothelial glycocalyx, increasing its thickness and decreasing stiffness, as a result of cystic fibrosis transmembrane conductance regulator (CFTR) activation [40]. CFTR regulates the transport of ions and water across the epithelial barrier. Therefore, it can be concluded that polyphenols decrease stiffness and increase hydration.

Higher flavonoid consumption has been associated with a lower arterial stiffness. Isoflavones and anthocyanins have been recommended as the most efficient for the improvement of vascular health [34]. Polyphenols activate endothelial nitric oxide synthase and increase endothelial synthesis and the production of nitric oxide [42], in such a way regulating arterial stiffness and preventing cardiovascular diseases. Anthocyanins are not only effective in the improvement of vascular health but also have antiviral properties. Inhibitory effects of anthocyanins [43] on different pathways involved in the virus life cycle have also been recently reviewed.

In an in-vitro study, it was shown that bovine coronavirus infections can be neutralized by theaflavins extracted from black tea [44]. Ueda et al. [45] reported that tannins decreased viral infectivity against 12 different viruses, which were both enveloped and non-enveloped. Extracts from persimmon (Diospyros kaki), which contains 22% of a food supplement persimmon tannin, demonstrated the highest antiviral effect. The authors demonstrated that extracts from persimmon inhibited attachment of the virus to cells. Pretreatment or posttreatment of cells with the persimmon extracts before or after virus infection did not inhibit virus replication. It was suggested that the antiviral effect of persimmon tannin can be explained by possible influenza virus protein aggregation. However, inhibited attachment of the virus to cells can also be explained by relative cell de-stiffening by persimmon extract due to changes in the ratio of virus and cell stiffnesses. Similarly, virus de-stiffening decreases platelet adhesion and prevents thrombus formation [7]. It has been shown that polyphenols can act as plasticizers; for example, tea-polyphenol-treated skin collagen demonstrated resistance to dehydration [46].

#### 2.4.2. Omega-3 Fatty Acids

A more hydrated and intact intervertebral disc tissue has been observed in rats who consumed an omega-3 fatty acids diet [47]. At the same time, omega-3 fatty acids have been reported to be a de-stiffening agent; long-term omega-3 fatty acid (fish oil) supplementation in a diet decreases arterial stiffness [48]. Fish oil supplementation decreases large arterial stiffness in overweight hypertensive patients [49]. Schmidt et al. analyzed the effect of omega-3 polyunsaturated fatty acids on the cytoskeleton and concluded that n-3 PUFAs regulated cytoskeleton-associated gene expression [50]. It has also been reported that Rho GTPase inhibited the metastatic ability of human prostate cancer cell line PC-3 by omega-3 polyunsaturated fatty acids [51]. It was also recently suggested that arachidonic acid, eicosapentaenoic acid (EPA), and docosahexaenoic acid (DHA) are able to inactivate enveloped viruses and increase resistance and recovery from SARS-CoV-2, SARS, and Middle East respiratory syndrome (MERS) infections [52]. Very low blood levels of the sum of eicosapentaenoic acid and docosahexaenoic acid were observed in USA, Italy, UK, and Brazil [53]. A high number of COVID-19 patients were reported in these countries.

#### 2.4.3. Drugs

Arterial wall stiffness is related to cardiovascular diseases [54]. For example, platelet adhesion and thrombus formation depend on arterial wall stiffness and hydration [7,55], Figure 1. An increase in substrate stiffness leads to endothelial monolayer disruption by increased cellular traction stresses and to increased endothelium permeability by leukocytes resulting in the development of atherosclerosis. It has been suggested that simvastatin can reduce matrix stiffness due to the decrease in RhoA activity in regulating contractility of the actin cytoskeleton and decrease in cellular traction forces. Simvastatin treatment resulted in improved endothelial barrier integrity and decreased endothelium permeability by decreasing cell–cell junction size in endothelial monolayers. It has been suggested that statins are able to maintain endothelial barrier integrity and prevent the development of atherosclerosis as a result of leukocyte penetration through endothelial monolayer damaged by tissue stiffness [56]. Similarly, it is possible that coronavirus penetration in novel COVID-19 disease can be increased for older patients due to increased age-related tissue stiffness. It was already reported that statins may decrease the fatality rate of MERS infection [57] and may be used in the treatment of COVID-19 [58]. The activities of various statins—simvastatin, atorvastatin, rusovastatin, simvastatin, lovastatin, mevastatin—and the possibility to use statins in treatment of COVID-19 was recently suggested [59].

Antihypertensive drugs, such as angiotensin-converting enzyme inhibitors, have been suggested to decrease arterial stiffness [60,61]. The possibility of using angiotensin-converting enzyme inhibitors in COVID-19 treatment must be investigated in more detail.

#### 2.4.4. Chitosan and Its Derivatives

It can be expected that AGE content will increase in the presence of high-molecular-weight chitosan [62]. However, chitosan oligosaccharides demonstrated angiotensin-converting enzyme inhibition activity [63]. It has also been reported that substitution of the hydrogen atom at the C-6 position of pyranose residue by the aminoethyl group promotes angiotensin-converting enzyme inhibitory effects of chitooligosaccharides [64]. It is known that angiotensin-converting enzyme inhibition dilates the blood vessels and decreases high blood pressure. Angiotensin-converting enzyme inhibitors can also be used to decrease arterial stiffness. Aminoethyl-chitooligosaccharide also suppresses proliferation of human lung A549 cancer cells [65]. Chitosan oligosaccharides suppress pathogenic microorganisms’ adhesion to cells and in such a way can be considered inhibitors of initial stages of infection processes [66]. Chitosan oligosaccharides are also able to penetrate inside pathogenic organisms and prevent their reproduction.

*N*-(2-hydroxypropyl)-3-trimethylammonium chitosan chloride (HTCC) was proposed as an inhibitor of human coronavirus HCoV-NL63. It was suggested that HTCC blocks human coronavirus interaction with the cellular receptor, angiotensin-converting enzyme type 2 (ACE2) proteins [67].

#### 2.4.5. Vitamins

Association between vitamin D deficiency and arterial stiffness determined from pulse wave velocity measurements was reported in a number of publications. Vitamin D deficiency was associated with the increase in arterial stiffness in children with type 1 and type 2 diabetes and with chronical kidney disease as well as in the elderly population [68,69,70,71,72,73,74,75].

Generally, the results from randomized controlled trials have been inconsistent and more large population studies should be conducted. It was reported that high-dose—but not low-dose—vitamin D supplementation was able to decrease arterial stiffness [74]. The decrease in arterial stiffness was related to the suggestion that 25-hydroxyvitamin D inhibits macrophage stimulation [76] and with the suppression of endothelin-induced vascular smooth muscle cell proliferation [77].

It has been recently reported that vitamin D supplementation could reduce the risk of death from the current COVID-19 epidemic. It is commonly accepted that vitamin D deficiency weakens the immune system. Vitamin D deficiency also increases tissue stiffness, and maintaining tight junctions was mentioned as one of mechanisms of vitamin D function [78]. Maintaining tight junctions prevents coronavirus infection by preventing tissue permeability.

Vitamin D intake is very low in Italy, Spain and France [79]. The most severe COVID-19 epidemic was observed in these countries, although, at present, there are not enough data confirming the link between vitamin D and COVID-19.

Vitamin E prevented an increase in femoral artery stiffness in diabetic Wistar rats [80]. It was reported that consumption of vitamin E for 2 months resulted in a decrease in carotid femoral pulse wave velocity and augmentation index in a group of 36 healthy men [81]. Arterial stiffness decreased as a result of long-term treatment with vitamin E in combination with vitamin C, coenzyme Q10, and selenium [82]. The effect of vitamin E in combination with pravastatin and hymocysteine on arterial stiffness was studied with the aim to reduce increased aortic and carotid artery stiffness in patients with end-stage renal disease [83].

It was suggested that supplementation with vitamin D and vitamin E may increase resistance to novel coronavirus SARS-CoV-2 [84].

Vitamin C decreases arterial stiffness [85,86]. A combination of vitamin C with various other drugs could be beneficial for COVID-19 patients [87].

Vitamin K2 decreases arterial stiffness by preventing and reversing calcification of arteries and inhibition of inflammation in the vascular wall [72,88,89,90].

It was reported that chitosan oligosaccharide ascorbate corrected deficiency in vitamin A, E, C and B groups [66].

#### 2.4.6. Salt Restriction

It was observed that salt consumption increases arterial stiffness and the risk of cardiovascular diseases [91,92,93,94,95], and salt also increases tightly bound water mobility due to the decrease in binding energy in pork meat [96]. Therefore, it is possible to suggest a correlation between arterial stiffness and increased tightly bound water relaxation time (increased protein mobility) measured by low field nuclear magnetic resonance.

Optimal hydration retards development of age-related diseases [97]. Increase in extracellular sodium concentration results in vascular endothelial cells’ activation and leads to shrinking and increased stiffness of glycocalyx [98,99,100]. Dehydration and related increases in sodium concentration also lead to the expression of inflammation mediators vascular cell adhesion molecule 1 (VCAM-1), endothelial-leukocyte adhesion molecule 1 (E-selectin), and monocyte chemoattractant protein 1 (MCP-1) [101,102]. High sodium concentration decreases NO release and promotes endothelial release of the pro-inflammatory cytokines IL-1ß and TNFα [103]. Increased endothelial stiffening and diminished glycocalyx coverage were observed in diabetic mice if compared with control. Glycocalyx coverage decreases in advanced age with diabetes progression [104].

#### 2.4.7. Dehydration, Stiffness and NF-κB Activation

Dehydration activates NF-κB [105]. Dehydration also results in enhanced tissue stiffness. Ishihara et al. reported that increased substrate stiffness leads to expression of inflammatory genes via NF-κB activation and results in actomyosin contractions, which triggers NF-κB activation [106]. The actomyosin contractions are induced by phosphorylation of myosin regulatory light chain (MRLC). The inhibition of MRLC phosphorylation by Rho kinase inhibitor Y27632 reduced the activity of NF-Κb. NF-κB activation may be associated with numerous diseases, including cancer and fibrosis development.

Matrix stiffening promotes F-actin polymerization regulated by small GTPase RhoA, Figure 2, and α smooth muscle actin (α-SMA)-containing stress-fiber formation. Inhibition of RhoA inhibits stress-fiber formation. Water molecules have an important role in F-actin polymerization, and it was shown that fewer water molecules were associated with actin when it polymerized [107]. Thus, the polymerization of actin and dehydration can be associated with an increase in tissue stiffness. The mechanical properties of actin can be regulated using various agents. For example, jasplakinolide, cytochalasin D, and latrunculin A were used to directly modulate actin polymerization of cultured rat airway smooth muscle cells [108].

Rho GTPases control cell proliferation and cytoskeleton remodeling and they are commonly hyper-expressed in tumors. Y-27632 also suppresses formation of stress fibers in cultured cells and decreases hypertension in several hypertensive rat models [109,110,111]. Pretreatment with ibuprofen was reported to prevent F-actin upregulation in articular chondrocytes from New Zealand White rabbits due to inhibition of RhoA pathway [112]. In the treatment of COVID-19, a number of doctors recommend using paracetamol rather than ibuprofen [113]. ROCK has been suggested as a potential target for the treatment of a number of diseases, such as cancer, neuronal degeneration, vascular diseases, kidney failure, asthma, acute lung injury, glaucoma, osteoporosis, erectile dysfunction, and insulin resistance [114,115,116,117,118,119]. For example, it has been recently reported that, in the treatment of acute lung injury, the Rho kinase inhibitor decreases inflammation, immune cell migration, apoptosis, coagulation, contraction, and cell adhesion in pulmonary endothelial cells. It means that all the above-mentioned diseases may be dependent on tissue stiffness related to time-dependent interaction of water molecules and tissue macromolecules. F-Actin reorganization is related to lung inflammation via increased blood neutrophil adhesion and migration. Pretreatment by cytochalasin B (CB) can block F-actin reorganization [120]. Currently, it is generally recognized that many questions remain to be investigated in order to understand Rho GTPase function, and it would be beneficial if tissue stiffness and the content of bound water were taken into account.

### 2.5. COVID-19 Age Dependence

It is very important that children have an extremely low rate of hospitalization presented by the U.S. Department of Health and Human Services in Figure 3. This fact demonstrates that people with soft tissues have a low chance of becoming ill with both age-related diseases and COVID-19.

It has been also reported that obesity as an underlying condition was observed in 48.3% COVID-19 patients. It was earlier demonstrated that obesity results in increased arterial stiffness [121].

## 3. Conclusions and Future Perspectives

Strong dependence on a patient’s age allows us to conclude that current methods of treatment of age-related diseases have to be verified, as well for application in the treatment of COVID-19. Thus, COVID-19 is unique and differs from previous epidemics by the fact that children are very rarely infected by novel coronavirus (Figure 3). COVID-19 virus particles interact with stiffer cell surfaces in old people, and, in cardiovascular diseases, platelets interact with stiffer arterial wall surfaces in old people. In both cases, stiffer substrate leads to a much more severe disease because, in both cases, the tissue surface adhesion and permeability increases. It is well known that children have the highest level of tissue hydration that decreases with advanced age and that tissue stiffness increases with aging. The hospitalization rate is very low for children having soft tissue and the hospitalization rate is high for the older population with hard tissue. Hence, there is an urgent need for the development of de-stiffening therapies.

Changes in water status lead to the change in extracellular matrix stiffness. The age-related decrease in tightly bound water content due to reduction of binding forces between water molecules and biological macromolecules results in the development of age-related and protein-conformational diseases due to an increase in tissue stiffness and permeability. It has been recently demonstrated that the ratio of extracellular water/intracellular water increases with advanced age and especially accelerates after the age of 70 years; a high value of this ratio is a major risk indicator for all-cause mortality.

The role of bound water in biological processes is underestimated in the current state of the art. It is evident that taking into account the effect of bound water on protein conformations and stability can facilitate understanding mechanisms of biophysical processes in age-related diseases and protein-conformational diseases. It is more correct to say that bound water is essential to life than to say that liquid water is essential to life.

Release of tightly bound water triggers glycation reactions that have a very important role in the aging process. Correlation between arterial stiffness and tightly bound water mobility can be expected. An increase in arterial stiffness is commonly accompanied by a decrease in glycocalyx coverage.

The relationship of bound water with tissue stiffness has scarcely been studied for the time being and can be suggested for in-depth study in future.

Biochemical glycation reactions and biophysical changes in hydration status are equally important in the development of physiological processes during aging. The possibility to regulate the hydration status of extracellular matrix could help in the development of new, more effective therapeutics for treatment of age-related and protein-conformational diseases.

NF-κB activation was associated with matrix stiffening and the development of age-related diseases. Rho GTPases control cell proliferation and cytoskeleton remodeling, and the inhibition of RhoA/ROCK inhibits stress fiber formation and has the potential to decrease the stiffness of extracellular matrix and, in such a way, to treat a wide number of pathologies. In order to understand Rho GTPase inhibitors’ functional properties more clearly, it would be reasonable—in parallel with tissue stiffness—to study changes in the content of tightly bound water using nuclear magnetic resonance.

Polyphenols, omega-3 fatty acids, vitamins D, E, C and K2, and statins have been suggested as de-stiffening agents. Research on tightly bound water’s role in de-stiffening is still in its infancy and needs a lot more attention. This review is focused on the role of dehydration and stiffness development in aging tissues. Evidently, other processes—for example, those involved in water–salt metabolism—have an important role in aging, and they could be discussed in special reviews.

## Figures and Tables

**Figure 1 polymers-12-01362-f001:**
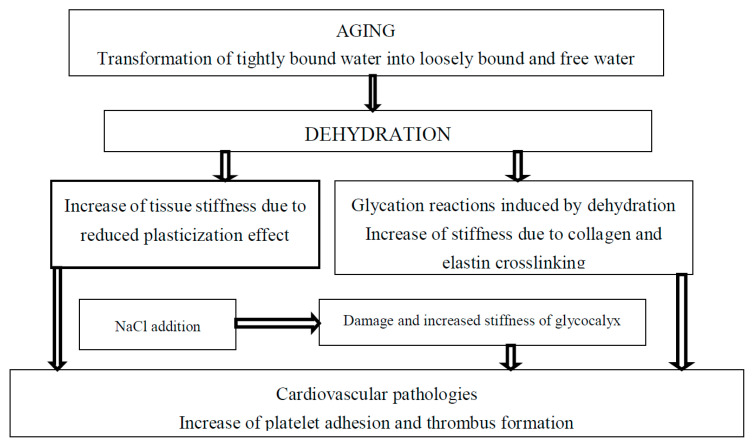
The effect of tightly bound water transformations on tissue stiffness and related cardiovascular diseases.

**Figure 2 polymers-12-01362-f002:**
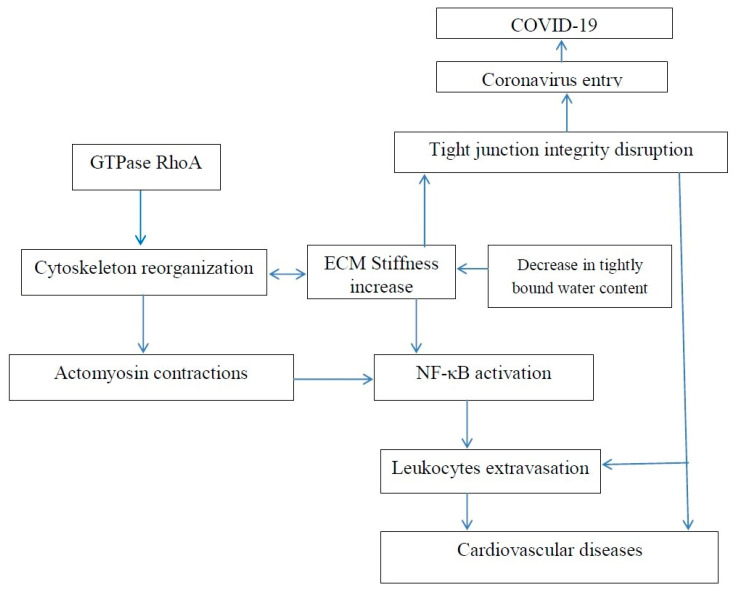
Increase in extracellular matrix stiffness results in both an increased risk of age-related cardiovascular disease and an increased risk of COVID-19.

**Figure 3 polymers-12-01362-f003:**
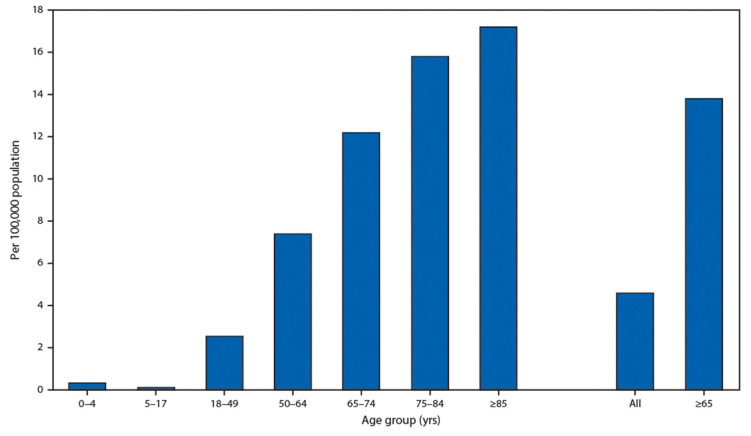
Coronavirus disease 2019 (COVID-19)-associated hospitalization rates, by age group, COVID-NET, 14 states, 1–28 March 2020. Cited from Garg S., Kim L., Whitaker M., et al. Hospitalization Rates and Characteristics of Patients Hospitalized with Laboratory-Confirmed Coronavirus Disease 2019, COVID-NET, 14 States, 1–30 March 2020. Morb Mortal Wkly Rep (MMWR) 2020; 69:458–464. doi:10.15585/mmwr.mm6915e3 *MMWR* and Morbidity and Mortality Weekly Report are service marks of the U.S. Department of Health and Human Services.

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
