# Peer review of "Role of Changes in State of Bound Water and Tissue Stiffness in Development of Age-Related Diseases"

_polymers, 2020, doi:10.3390/polym12061362_

Round 1
Reviewer 1 Report
Polymers. Manuscript 782422
Title: Role of Changes in State of Bound Water and Tissue 2 Stiffness in Development of Age-related Diseases
This manuscript proposes on review on bound water and tissue stiffness in age-related diseases. The subject is of interest and novel, and good writing, however the paper still suffers some limits and the author should try to explain the possible mechanisms by which bound water contributes to age-related diseases more clearly and organized.
Major comments:
- Roles of bound water and tissue stiffness in age-related diseases still need to be fully described and discussed.
- The paper provides a lot of research data on de-stiffening materials, mainly about their normal functions, but seldom about their relationship with bound water and stiffness, which needs to be further discussed.
- The structure of the article had better be adjusted to a more logical order, especially the part of “De-stiffening”.
- The figures in this paper need to be modified to a good appearance.
- In conclusions, the viewpoints are scattered, repeating the contents.
Minor comments:
- Line60-61, “for example, in hyaluronic acid hydrogels the differentiation behavior of human mesenchymal stem cells (hMSCs) depends on cellular traction”, more details and direct relations to the stiffness should be discussed in the reference.
- Paragraph “2.1. Tissue Stiffness”, It's better to switch the order of the last two sentences, references 11 and 15. That will be more logical in the sentences.
- Line 79-88, “2.2. Tightly Bound Water”, it is necessary to explain the correlations of the ICW, ECW and the tightly bound water.
- Line 113, “Polyphenols increase synthesis and production of nitric oxide [42].” This sentence is isolated and lacks contexts.
- Line 114, “Inhibitory effects of anthocyanins [43] on different pathways involved in virus life cycle have also been recently reviewed.” This sentence, similar problem as “4”, is isolated and lacks context.
- Line 116, “in vitro study”, “in vitro” should be in italics.
- Line116-127, in this paragraph, it is necessary to explain more detail data about the functions of persimmon tannin inhibited attachment of the virus to cells by cells de-stiffening.
- Line153, “as a result of leukocytes penetration through damaged by tissue stiffness endothelialmonolayer”, please explain the sentence more clearly.
- Line 162, “A wide number of COVID-19 potential drugs have been recently overviewed”. This sentence has no relation with the contents.
- Line 163-172, “Chitosan and its Derivatives”, what is the correlation with the stiffness? Please make a supplement and describe clearly.
- Line 188, “Vitamin D intake is very low in Italy, Spain and France [79]. The most severe COVID-19 epidemic was observed in these countries. ” This sentence suggests a causal link between vitamin D and COVID 19, but there is not enough data.
- Line 203-205, it is better to put the mechanism of chitosan oligosaccharides in the previous paragraph 2.4.4, for it is no evidence to show the anti-infection functions are related to the vitamins.
- Line 259, “2.5. Statistics”, this subtitle is not appropriate.
- There are some problems with verb tense, boldface.
- Some literature formats are not uniform.
Author Response
Thank you for your comments. The responses to comments and files with revised version are attached

Reviewer 2 Report
The presented manuscript is a review article discussing Role of changes in state of bound water and tissue stiffness in development of age-related diseases. The text of the article is original.
The results of the cited papers are interpreted adequately. The manuscript raises an important question about the aging and the development of age-dependent diseases. More than 100 scientific articles are cited in the manuscript, more than half of which have been published over the past 5 years. Accordingly, the author raises an urgent question about the role of dehydration and increasing tissue stiffness and tries to find the solutions to the problem of eliminating the causes of age-dependent diseases.
The part of the article that discusses the state of water molecules in different tissues is interesting and important. This aspect of the problem is really little studied and its emphasis in this article is an important element. However, consideration of this problem is presented only in short text with a few links. But, this is understandable, since so far this field has been poorly studied.
Despite the fact that the article analyzes the data of recent years, the problem of increasing dehydration with age is not a new one. But since it has still not been completely resolved, the review article presented, of course, is relevant.
Broad comments
The basis of the article is a consideration of possible factors that reduce tissue dehydration and stiffness. However, it should be noted that this part of the review is very incomplete. On the other hand, it must be recognized that it is very difficult to make it complete, since it is impossible to list all the factors involved in the development of age-dependent diseases, including those that control the body’s water-salt balance, in one review article. To date, a huge number of factors and genes are known to be involved in the development of age-related diseases, including those involved in water-salt metabolism. However, the author does not even mention the influence of hormonal systems that regulate the water and water-salt balance.
Perhaps the author is right in emphasizing tissue dehydration and stiffness as the basis for the development of many age-related diseases, but it should be at least noted in the text that the factors listed in the article are far from a complete list of substances whose balance change can affect the hydration/dehydration of tissues and the development of age-dependent diseases.
New hypotheses presented in this review article include the author’s assumption that COVID-19 is one of age-dependent diseases, the sensitivity to which increases with decreasing hydration, and, accordingly, increasing tissue stiffness with aging. Despite the fact that the author is trying to build some logic, based on disease statistics and currently used drugs that are recommended for the treatment of COVID-19 and are used in the treatment of age-related pathologies, it is difficult to agree with this hypothesis, since COVID-19 is infectious disease, in contrast to hypertension, diabetes, neurodegenerative and other diseases, which are currently considered to be age-dependent.
Of course, people with age-related diseases have an increase in tissue stiffness and a decrease in immunity. However, the author, while proposing the physiological hypothesis of hypersensitivity of aged people to COVID-19 infection, does not at all mention that the virus has affinity for targets (expressed proteins, the expression of which is associated with the pharmacological effects of a number of drugs used in the development of age-dependent diseases (for example, when using antihypertensive drugs , which, quite naturally, are more often used by older people).
Offering his vision of the foundations of the development of COVID-19, the author writes that “Antihypertensive drugs, such as angiotensin-converting enzyme inhibitors, have been suggested to decrease arterial stiffness” (lines 160-161), which the author believes is very useful for combating coronavirus, however data that the use of angiotensin-converting enzyme inhibitors leads to an increase in the number of its targets (ACE2) in the cells, which makes patients more vulnerable to the development of COVID-19, the author does not discuss.
Specific comments
- Some phrases inserted into the text are not relevant to the topic under discussion. For example, Line 30. Dehydration during freezing except in this phrase is no longer mentioned in the article.
- In general, the English in the presented manuscript is appropriate and understandable, but in some places text correction is required (for example, lines 28-29).
Author Response
Thank you for your comments. The file with responses to comments is attached and required additions to the text and correction have been made

Round 2
Reviewer 2 Report
Comments on the manuscript are in the attached file.

Author Response
The requested corrections have been made. Please see the attachment
